# An Exploratory EEG Analysis on the Effects of Virtual Reality in People with Neuropathic Pain Following Spinal Cord Injury

**DOI:** 10.3390/s22072629

**Published:** 2022-03-29

**Authors:** Yvonne Tran, Philip Austin, Charles Lo, Ashley Craig, James W. Middleton, Paul J. Wrigley, Philip Siddall

**Affiliations:** 1Department of Linguistics, Macquarie University Hearing, Macquarie University, Sydney, NSW 2109, Australia; 2Department of Pain Management, HammondCare, Greenwich Hospital Greenwich, Sydney, NSW 2065, Australia; paustin@hammond.com.au (P.A.); phil.siddall@sydney.edu.au (P.S.); 3Management Disciplinary Group, Wentworth Institute of Higher Education, Surrey Hills, NSW 2010, Australia; charles.lo@win.edu.au; 4Sydney Medical School-Northern, Faculty of Medicine and Health, The University of Sydney, Sydney, NSW 2006, Australia; a.craig@sydney.edu.au (A.C.); james.middleton@sydney.edu.au (J.W.M.); paul.wrigley@sydney.edu.au (P.J.W.); 5John Walsh Centre for Rehabilitation Research, Kolling Institute, Northern Sydney Local Health District, St Leonards, NSW 2065, Australia; 6Pain Management Research Institute, Kolling Institute, Northern Sydney Local Health District, St Leonards, NSW 2065, Australia

**Keywords:** EEG, brain activity, virtual reality, neuropathic pain, spinal cord injury, fractal dimension

## Abstract

Neuropathic pain in people with spinal cord injury is thought to be due to altered central neuronal activity. A novel therapeutic intervention using virtual reality (VR) head-mounted devices was investigated in this study for pain relief. Given the potential links to neuronal activity, the aim of the current study was to determine whether use of VR was associated with corresponding changes in electroencephalography (EEG) patterns linked to the presence of neuropathic pain. Using a within-subject, randomised cross-over pilot trial, we compared EEG activity for three conditions: no task eyes open state, 2D screen task and 3D VR task. We found an increase in delta activity in frontal regions for 3D VR with a decrease in theta activity. There was also a consistent decrease in relative alpha band (8–12 Hz) and an increase in low gamma (30–45 Hz) power during 2D screen and 3D VR corresponding, with reduced self-reported pain. Using the nonlinear and non-oscillatory method of extracting fractal dimensions, we found increases in brain complexity during 2D screen and 3D VR. We successfully classified the 3D VR condition from 2D screen and eyes opened no task conditions with an overall accuracy of 80.3%. The findings in this study have implications for using VR applications as a therapeutic intervention for neuropathic pain in people with spinal cord injury.

## 1. Introduction

Spinal cord injury (SCI) is a life-changing event that causes not only a debilitating loss of sensorimotor and autonomic functions but is also associated with numerous secondary conditions. One prevalent secondary condition is chronic pain, with research showing over 50% of patients reporting more than one pain type and the pain often described as unrelenting and excruciating [1,2]. For people with SCI, neuropathic pain (NP) has been reported to be as common as musculoskeletal pain [2]. At the injury level, SCI NP is thought to result from altered central neuronal activity, with hyperexcitable neurones having exaggerated responses to stimuli; however, below the level of injury, the mechanisms are less clear [3]. The neurophysiological responses are thought to generate abnormal pain impulses back to the brain. SCI also leads to the reorganisation of the primary somatosensory cortex, which is associated with abnormal patterns of firing in the cortex and thalamus, known as thalamocortical dysrhythmia (TCD) [4], and is proposed as a mechanism underlying the generation of neuropathic pain and other neurological symptoms [5,6].

Given these complex mechanisms, involving structural and functional changes in central pain pathways at multiple levels of the neuroaxis, current treatments provide only partial and often unsatisfactory pain relief [7]. As such, alternative therapeutic approaches, such as virtual reality (VR), are now being examined [8] where advancement in technology offers an alternative treatment for a number of medical and psychological conditions and procedures [9,10,11]. VR is a simulated creation of a 3D environment using computer technology [12]. Current VR systems include head-mounted devices (HMD) with 3D-enabled glasses, noise-cancelling headphones for sound and head and/or body-tracking sensors in addition to devices such as joysticks and data gloves [13]. Together, this forms a realistic multisensory experience that surrounds the user, generating strong feelings of “presence”, a subjective sensation of being in another place [14].

Several pilot studies using a variety of 3D HMD and 2D screen-based VR applications have shown a reduction in NP in people with SCI pain in over two-thirds of participants [15,16,17]. Such encouraging findings suggest that VR may be an effective, accessible, and inexpensive method of reducing NP in both the long and short term. Recent evidence suggests that, compared to 2D VR, 3D VR technologies are more realistic and vivid [18], where the three-dimensional perception of an image or video is considered more immersive where users feel completely involved [19]. 

Although clinical studies in people with SCI-related NP have shown promise for the effectiveness of VR, the neural mechanism underlying the positive response to VR is unknown. In previous studies, there is evidence for neural mechanisms underlying VR immersion. From electroencephalography (EEG) studies, task-related differences in EEG alpha activity and coherence were correlated with spatial presence [20]. Frontal-midline theta activity increases were found from different levels of immersion in VR applications [21].

Given that the brain activity of SCI people with NP has been found to be associated with resting-state EEG [6,22,23], we were interested in examining underlying brain activity changes in persons with SCI and NP during VR intervention. Current studies have demonstrated brain activity markers for NP, specifically, increases in theta- and beta-wave frequencies and reduced alpha-wave frequencies in EEG signals, thought to be associated with TCD [6,22,23]. Another study from Vuckovic and colleagues showed that these EEG frequency changes can be used to identify patients with SCI who are at risk of developing NP before physical symptoms appear [24]. These EEG markers of NP have also been shown to be reversible following treatments to reduce SCI pain, where, for example, Hasan and colleagues showed significant reductions in beta- and theta-wave frequencies following biofeedback treatment [22]. Additionally, recent pilot studies investigating EEG and VR encouragingly show (a) decreases in beta-wave frequencies in response to VR in people with anxiety [23], and (b) in a case report, increases in alpha-wave frequency during phantom limb pain relief in people with brachial plexus injury during VR [25]. 

Thus, the aim of the current study was to determine whether use of VR is associated with corresponding changes in EEG patterns linked to the presence of neuropathic pain. We hypothesised that using a 3D VR application would be associated with a shift of EEG activity from a TCD brain wave pattern towards a reduced TCD state and thus a reduction in the severity of NP. We examined brain activity in three states, a resting eyes-open state with no task (EO-no task), using a 2D screen-based VR (2D screen) as an active control and during immersive 3D HMD VR (3D VR). 

## 2. Materials and Methods

### 2.1. Study Design

We used a randomised cross-over study design for this exploratory study. This involved two sequential VR interventions and a baseline measure using within participant comparisons. Baseline measure was taken for EEG comparisons involving an eyes-open condition, whereby participants were asked to remain still and focus on the middle of a blank computer screen. There were two VR interventions, one utilising an immersive 3D VR and one with 2D screen applications using the same virtual environment. Seventeen adults with SCI and known NP were recruited using convenience sampling. We randomly allocated the type of VR intervention used first and second using sequentially numbered, opaque sealed envelopes. As it was important to show parity in describing both interventions, a script using neutral language was prepared. This study was registered by the Australia New Zealand Clinical Trials Registry, number ACTRN12618000959279, in May 2018, and further detail on the exploratory trial can be found in Austin et al., 2020 [26].

### 2.2. Participants

Participants were recruited from both a database of participants with SCI as well as through clinical contact. The inclusion criteria were adult males with SCI of longer than 12 months duration, lesion at C6 level or below, a confirmed diagnosis of NP (>6 months), reported neuropathic pain over the previous week prior to attending interventions, and stable pharmacological or no pharmacological treatment for at least four weeks. We limited the study to male participants only as they account for the majority of new SCI cases (up to 80%) and because of potential gender differences in pain reporting and medication use [27]. The exclusion criteria were the presence of other pain types that were more prominent during the time of the interventions, a SCI level higher than C6, presence of brain injury, or other neurological diagnosis. 

### 2.3. Study Schedule

All participants attended the intervention on one occasion. To account for circadian influences on wakefulness in the brain activity of people with SCI, all participants were asked to attend at 11 a.m. Baseline pain intensity measures were taken with an 11-point numerical pain rating scale (NPRS). We examined average, worst, least and current NP intensities. Current NP intensity were taken immediately after the intervention and used for the analysis. As we used a cross-over design, we included a washout period in the experimental design. This was implemented to reduce any potential carryover effect that may be from the effects of the first intervention. The washout period separated the two intervention periods. Washout periods need to be at least five times the half-life of a given treatment [28], so a 60 min washout period was used. The hour-long washout was calculated from reports that pain significantly reduces immediately after VR exposures but returns to baseline levels at 10 min after VR exposure [29]. The cross-over was counterbalanced so that exposure to both interventions were equal. The entire study took place in a temperature-controlled room maintained at 25 °C. Details on the intervention protocol can be found in Austin et al. (2020) [26]. The height of the bench for the screens was modified for wheelchair access and adjusted appropriately for each person. Participants were required to report any headset discomfort and cyber-sickness (includes symptoms of nausea, vomiting, headache, vertigo and fatigue) prior to, during or after using the 3D HMD VR device. 

### 2.4. 3D HDM VR Device and Task

The Oculus Rift^®^ headset is commercially available, inexpensive and commonly used for VR studies in medical research [30]. The screen sampling rate was 80 Hz. For the VR task in this study, participants viewed a 3D VR experience called Nature Trek^®^ that includes nine nature environments all containing many types of animals and calming music. Prior to use, participants were instructed on the use of a hand-held joystick to move around an alpine meadow environment and make full use of the 360° scene. The VR application was standardised across the group. The VR headset was calibrated for participants’ eyesight in addition to advice on motion sickness prevention during VR such as reducing the speed of their character and/or reducing head movement. 

### 2.5. 2D Screen Application 

The same Nature Trek^®^ application was run on a 17.3-inch Alienware^®^ laptop screen with the participant seated in the same position. This allowed for a reliable comparison between the effects of 3D VR and 2D screen experiences. The screen sampling rate was 60 Hz.

### 2.6. Self-Reported Pain Measures

We used the numerical pain rating scale (NPRS) to investigate the effects of 3D HMD VR and 2D screen applications on SCI NP. The NPRS was completed at three time points to gather pain information for baseline and VR interventions. Participants completed the 11-point NPRS after each intervention and reported levels of pain intensity immediately after the intervention, as well as reporting their average pain intensity during each intervention and lowest pain intensity during each intervention. The 11-point NPRS is a reliable and valid measure used across many pain populations [31]. Changes in pain intensity from these VR interventions has been reported in our previous feasibility study (see Austin et al. [26]).

### 2.7. EEG Recording and Preprocessing 

Thirty-two EEG channels were measured using the EmotivPro^®^ system over the entire cortex, following the International 10–20 Montage System. EEG was recorded using the EmotivPRO^®^ software and was sampled at a 256 Hz sample rate with left and right ear (A1, A2) references. Once fitted, the Oculus Rift VR HMD system was placed over the top of the EEG cap (Figure 1). Two minutes of baseline EEG was taken. During baseline, participants were instructed to remain still and focus on the middle of a blank computer screen to avoid eye movement. They were asked to fixate on a printed cross placed in the middle of the screen. The VR interventions were each 15 min in length.

EEG pre-processing was conducted in the following steps.
The EEG signals were re-sampled to 128 Hz from EmotivePRO^®^EEG signals were imported into EEGLabAll channels were transformed to the average reference in EEGlab.EEG signals were filtered with a 0.1 Hz high-pass filter.EEG signals were visually inspected so that 20 s segments relatively free of major artifact could be extracted from the three conditions EO-no task, 2D screen and 3D VR. EEG from EO-no task was taken at approximately the 1 min point, after participant’s EEG had settled and both 2D screen and 3D VR were taken at approximately the 5 min mark during the immersive task, where the first 20 s of relatively clean EEG segment could be obtained.With the 20 s EEG segment, EEG artifact from blinks, eye saccades, lateral eye movements and cardiac signal components were analysed using independent component analysis (ICA) using EEGLab [32]. Artifact components were visually inspected and manually removed. Fast frequency and EMG (electromyography) noise was then removed using the Automatic artifact removal (AAR) plugin for EEGLab [33].

### 2.8. Spectral Analysis

To determine the corresponding changes in EEG patterns linked to the use of VR in those with neuropathic pain, we started with an examination of EEG spectral activity using a robust power spectral estimation method following Melman and Victor (2016), which utilises a multi-taper method [34]. We used this method to ensure that any noise from the VR system does not influence the spectral EEG activity, as it is more resistant to transient artifacts. Spectral analysis quantifies the amount of oscillatory activity in the different frequencies, and we examined the relative power for the widely accepted frequency bands: the delta (1–4 Hz), theta (4–8 Hz), alpha (8–12 Hz), beta (12–30 Hz) and low gamma (30–45 Hz) waves. Relative power for the spectral bands was calculated as the power of each given band divided by the sum power from 1 to 45 Hz. To ensure that signals were not affected by the VR headsets (that is, electrode sites where the VR device band sat on top) and representative of the regions of the scalp, we chose nine channels that covered the frontal region (F3, Fz, F4), central region (C3, Cz, C4) and parietal region (P3, Pz, P4). The conditions were baseline eyes open (EO), 2D screen and 3D VR. 

### 2.9. Fractal Dimension Analysis

We also explored a nonlinear and non-oscillatory EEG feature using Fractal Dimension (FD). Whereas spectral analysis explored oscillatory markers and different frequency bands, using FD, we could examine non-oscillatory markers for the different immersion levels from VR interventions. The FD of an EEG signal measures its complexity, that is, the amount of irregularity within the time series. We explored the FD of data from individual EEG channels using a method used most with EEG signals that was introduced by Higuchi (1988) [35]. The Higuchi’s FD is a straightforward method that can be applied to time series data in order to extract the fractal dimension.

Suppose we have a time series:X(i)(i=1,…..N)

From this, the length of the curve $L_{m}(k)$, for m=1,….k can be defined as follows:$$L_{m}(k)=\frac{1}{k}\{(\sum_{i=1}^{\left[\frac{N-m}{k}\right]}\left|X(m+ik)-X(m+(i-1)k\right|)\frac{N-1}{k[\frac{N-m}{k}]}\}$$
where the square brackets [] denotes Gauss’ notation, both *m* and *k* are integers, *k* indicates the discrete time interval and m indicates the initial time value. 

The length of the curve for the time interval *k* is then defined as:$$L(k)=\frac{1}{k}\sum_{m=1}^{k}L_{m}(k)$$

If the curve contains fractal properties, then L(k) is proportional to $k^{-D}$, where *D* is the fractal dimension. The value of the time interval is varied from *k* = 1, 2, 3 up to $k_{\max}$. 

A log–log plot of L(k) against *k* will give a straight line with slope–*D*.

From numerical analysis, a choice of $k_{\max}$ = 6 was found to sufficiently estimate the slope. The FD for this study was calculated over time in a sliding non-overlapping window with a fixed length. 

### 2.10. Artificial Neural Network Analysis and Evaluation Metrics

We used the artificial neural network (ANN) analysis from the SPSS v27 toolbox (SPSS Inc., Chicago, IL, USA) to demonstrate whether neural differences existed in brain activity between three immersion conditions that are distinct and can be classified using ANN models. We used a multilayer perceptron (MLP) ANN model with three-layer feedforward back propagation. EEG spectral and FD data were randomly divided into training (70%) and testing (30%) sets. A hyperbolic tangent function was used for the hidden and output layer. A gradient descent was used to estimate the synaptic weights. The initial learning rate was set as 0.4 with a momentum of 0.9. We performed these models for binary classifications. For performance of the classifications, we used the receiver operating characteristics (ROC) analysis as a measure of predictive accuracy. We also used well-known performance indicators sensitivity or true positive rate (TPR), specificity or true negative rate (TNR) and accuracy, obtained from the testing sample. These were calculated as follows:$$Sensitivity=\frac{TP}{TP+FN}$$
$$Specificity=\frac{TN}{TN+FP}$$
$$Accuracy=\frac{TP+TN}{TP+TN+FP+FN}$$

## 3. Results

One participant was excluded from all analysis due to reporting no pain over the previous week before the study. Another participant was excluded from EEG analysis as they had poor and corrupted EEG signals. All participants reported no cybersickness following the VR interventions. Table 1 shows the participants’ demographic characteristics including age, duration in years since SCI, level and extent of SCI, pain consistency and prescribed pain medication.

### 3.1. Evidence of Improvement in Pain Scores from Participating in the Tasks

The mean pain intensity scores over the week prior to their attendance, during and after 2D screen and 3D VR interventions were examined. Repeated measures ANOVA showed overall significant differences in pain ratings for all three conditions (pre-task, 2D screen and 3D VR), F (2, 14) = 46.6, *p* < 0.001. Post hoc analysis using the Bonferroni test showed significant reductions within participants from pre-task to interventions (*p* < 0.001), with mean (95% CI) pain ratings of 4.9 (4.1–5.8) for pre-task, 3.4 (2.4–4.4) for 2D screen and 1.9 (1.0–2.9) for 3D VR.

### 3.2. Regional Differences in Relative Power for the Three Conditions

Repeated measures MANOVAs were conducted to test for differences in the spectral relative power of the three conditions for the three regions, frontal (F3, Fz, F4), central (C3, Cz, C4) and parietal (P3, Pz, P4). Table 2, Table 3, Table 4, Table 5 and Table 6 shows the relative power (Mean (SE)) breakdowns for each of the frequency bands from the nine sites in the three conditions. Univariate main effects from repeated-measures ANOVAs were used to determine any statistical differences between the three conditions. For the frontal region, EEG differences were found in the relative delta (Wilks’ Lambda = 0.52, F (6, 52) = 3.34, *p* = 0.007, η^2^_P_ = 0.28), theta (Wilks’ Lambda = 0.56, F (6, 52) = 2.92, *p* = 0.016, η^2^_P_ = 0.25 alpha (Wilks’ Lambda = 0.47, F (6, 52) = 4.0, *p* = 0.002, η^2^_P_ = 0.31) and gamma (Wilks’ Lambda = 0.58, F (6, 52) = 2.72, *p* = 0.023, η^2^_P_ = 0.24 frequencies. Post hoc test using Bonferroni found significant increases in delta activity in the F3 and F4 sites with the 3D VR condition. A significant reduction in frontal theta was found in the Fz site, whereas a reduction in alpha activity was found in F3 and F4, and was greatest during 3D VR. There were statistical differences in the beta band, but significant increases in gamma activity were found in the F3 site with greatest increase for the 2D screen condition. For the central region, relative power difference was found for the theta frequency only (Wilks’ Lambda = 0.57, F (6, 52) = 2.77, *p* = 0.020, η^2^_P_ = 0.24), with significant decreases for the VR interventions compared with resting EO. The alpha frequency band did not show an overall significant difference between the three conditions, despite significant differences in the univariate main effects. In the parietal region, relative power differences were significant for the alpha (Wilks’ Lambda= 0.58, F (6, 52) = 2.75, *p* = 0.021, η^2^_P_ = 0.24), theta (Wilks’ Lambda = 0.54, F (6, 52) = 3.09, *p* = 0.012, η^2^_P_ = 0.26) and gamma (Wilks’ Lambda = 0.46, F (6, 52) = 4.09, *p* = 0.002, η^2^_P_ = 0.32) frequency bands. There were significant reductions in relative theta and alpha power for the 3D VR condition. Gamma activity increases were also found to be significantly greater for the 2D screen condition.

Figure 2 shows the overall EEG power spectrum for the three conditions in three EEG channels Fz, Cz and Pz, representative of the frontal, central and parietal regions. An increase in delta activity occurred for the VR interventions compared with baseline EO condition. The greatest reduction in the theta band was observed for the 3D VR condition. The reduction in the alpha frequency band was gradual, with the greatest reduction during the 3D VR intervention. Increases at higher frequencies occurred in the gamma frequencies (30–45 Hz). This can be seen with 2D screen and 3D VR, with greater increases during the 2D screen task.

### 3.3. Regional Differences in Higuchi’s FD for the Three Conditions

Repeated measures MANOVA also was conducted to test for differences in FD for three regions, frontal (F3, Fz, F4), central (C3, Cz, C4) and parietal (P3, Pz, P4). There were no significant differences in the FD for the three conditions in the frontal region. There were significant FD differences in brain activity for both the central region (Wilks’ Lambda = 0.21, F (6, 9) = 5.6, *p* = 0.011) and the parietal region (Wilks’ Lambda = 0.27, F (6,9) = 4.1, *p* = 0.028). Post hoc analysis using Bonferonni found differences between the eyes-open with both 2D screen and 3D VR, but not between 2D screen and 3D VR. The mean percent change for 2D screen from baseline EO was 4.73%, and the mean percent change for 3D VR from baseline EO was 5.08% (See Table 7 for summary statistics). Figure 3 shows the FD by EEG channel for each of the three conditions. Compared with the eyes-open task, both 2D screen and 3D VR conditions displayed raised FD.

### 3.4. Performance of ANN model for Classifying 3D VR Using EEG Activity

The final three-layer model consisted of 6-6-2 feedforward back propagation. Table 8 shows the performance of the ANN model for each binary classification explored. For the classification of 3D VR against both EO-no task and 2D screen, we obtained an overall accuracy of 80.3%. However, sensitivity for this model was low at only 43.8%. The highest sensitivity was between 3D VR with EO-no task at 78%. The differences in the sensitivity, specificity and accuracy demonstrate that the neural activity during 3D VR was distinct and can be classified.

## 4. Discussion

The occurrence of NP in people with a SCI is thought to have a neural basis that is both complex and multilevel. Current available treatments have only provided partial and often unsatisfactory pain relief [36]. In this study, we explored a novel alternative therapeutic approach of utilising VR to reduce NP in people with SCI while examining the underlying brain activity during the interventions. As reported in our previous paper [26], our findings showed a significant reduction in self-reported pain intensity ratings in participants with SCI for their NP during VR interventions. The reduction in pain intensity was greatest during the 3D VR task. This demonstrates that VR interventions can be viable alternative therapeutic interventions for NP in persons with SCI.

We then examined whether there were any corresponding neural changes during the VR intervention. The results from this study found a reduction in EEG theta in the frontal and parietal regions. The decrease in relative theta was greatest for the 3D VR intervention. A significant and consistent reduction in relative EEG alpha frequency band was found in almost all EEG sites for 3D VR and 2D screen. The full EEG spectrum averaged for all participants shows the reduction to be gradual in the frontal and parietal regions and based on the level of immersion, such that reduced alpha power was greatest during 3D VR. These EEG changes for the three conditions partially supports our hypothesis that EEG changes will shift in the direction towards reduced TCD with decreased theta activity and increased alpha activity. A reduction in theta activity was observed during VR interventions; however, rather than an increase in alpha activity to counter the TCD, we found a further decrease in alpha frequency power during VR application. Similarly, Jensen et al. (2013) confirmed TCD EEG patterns in chronic pain for SCI (increased theta and reduced alpha), but they also found significant associations between pain severity and EEG alpha wave activity, with higher alpha activity associated with increased levels of pain [37]. They concluded that successful pain suppression may be associated with decreased frontal alpha activity, and this was confirmed in the current study.

The reduction in EEG alpha power, with increases in delta and low gamma activity, may also be associated with the “distraction” or immersion effects of VR applications with 3D VR thought to have the greatest degree of immersion [38]. Lim et al. 2019 found alpha waves to decrease during concentration and immersion [39]. Similarly, in a study with cancer pain and VR immersion, it was low frequency power in theta and alpha frequency ranges that were found to decrease during VR meditation task compared to their pre-condition [40]. Other EEG changes such as frontal delta activity increases has also been found to be associated with concentration during cognitive tasks [41], and it is thought that this link is moderated by motivation [42]. We found increases in frontal delta activity, and this only occurred during the 3D VR condition and not during the 2D screen. Low gamma activity increases were highest for the 2D screen condition in the frontal region but were the same as in the 3D VR in the parietal region. Gamma activity has often been linked to cognitive function or processes [43]. Increases in gamma activity are thought to be related to perception [44].

We were also interested in non-oscillatory EEG markers for NP and effects of VR interventions. FD is a nonlinear measure for complexity in brain signals. The FD of EEG signals have repeatedly been shown to be of a lower value for people suffering from brain disorders compared to healthy individuals [45]. Anderson and colleagues (2021) found able-bodied participants to have higher FD compared to SCI participants with neuropathic pain and used FD as a diagnostic marker for NP [45]. Foss and colleagues (2006) were able to differentiate between different pain states from FD values. They found FD to be lowest for thermal pain and greatest for back pain [46]. Using Higuchi’s FD, we found significant increases in FD for the central and parietal regions during 2D screen and 3D VR compared with the EO-no task. However, there were no distinguishable differences between the FD for 2D screen and 3D VR. The increase in FD for 2D screen and 3D VR may be showing changes in neural signals, demonstrating a normalisation of the affected thalamocortical system. Higher FD values generally correspond to higher signal complexity, and a reduction in FD may indicate a loss of neural efficiency, as previously found in people with Alzheimer’s disease [47,48].

Although we did not find significant differences for 2D screen and 3D VR in FD, we were able to distinguish the two brain activity states using ANN. Using both oscillatory and non-oscillatory measures for our feature set, classification of 2D screen against 3D VR had an accuracy of 68.3%. Classifying 3D VR from both 2D screen and EO-no task gave an overall accuracy of 80.3%; however, this was largely from the high-specificity (true negative) result. Sensitivity was only at 43.8%. This low sensitivity was probably due to the gradual changes from level of immersion between baseline EO to 2D VR to 3D VR, making it difficult to detect 3D VR from a mixture of immersion levels. Classifying 3DVR from baseline EO had highest predictive accuracy with an ROC of 0.877. The results match the findings from the spectral bands, in that the gradual changes in immersion levels are reflected with the sensitivity analysis. Sensitivity was highest between baseline EO and 3D VR. Sensitivity between baseline EO with 2D screen and 3D VR with 2D screen were similar at 66.7% and 68.6%, respectively. The results from the ANN models indicate that brain activity during 3D VR immersion is distinct and can be classified; the ROC shows reasonable predictive accuracy.

## 5. Strengths and Limitations

The findings in this study have implications for using VR applications as a therapeutic intervention for NP in people with SCI and our understanding of the mechanisms responsible for VR-associated pain relief. Both changes in alpha and theta wave activity have been demonstrated in association with SCI-related NP. The significant reduction in self-reported pain intensity after the intervention was found to correspond to significant changes in EEG brain activity. The changes seem to suggest that two pathways may be occurring during VR intervention. There is some evidence that VR-associated pain relief is associated with a remediation or reversal of TCD through reductions in theta activity, but there is also evidence for a possible attention-related mechanism involving alpha activity, delta activity in the frontal cortex, and low gamma activity in the parietal region. The strengths in this study include the use of both oscillatory and non-oscillatory methods in EEG signal processing to understand the underlying neural mechanisms during VR immersion. As to limitations, we applied a common average reference strategy for the EEG signals to first remove any common noise. However, common average referencing can introduce bias and lower amplitudes of the signal when the coverage of electrodes is not dense enough. Our study only utilized 32 channels, and the recommended electrode density is at least 64 channels (Nunez, 2006). As such, there may have been bias in the amplitudes of the signals in this study. Additionally, as this study is a preliminary exploratory examination, we did not examine post-VR session effects, and we are unable to determine if the effects on pain reduction are able to persist longer than the VR session. However, a more recent study using the same VR distraction protocols in people with cancer pain did show pain relief remained for up to 20 min after the VR sessions [49]. Encouragingly, recent work also shows that more frequent use of cognitive-based VR applications over weeks or months in people with NP and phantom limb pain in combination with other non-pharmacological therapies offers both long-term relief in pain intensity and decreases in pain-related behaviours [50,51]. Future research should be conducted with a larger sample with focus on longer-term outcomes to test for cumulative effects from VR interventions. A larger study should also examine different baseline pain intensity levels to test whether this intervention can be used in people with high levels of pain.

## Figures and Tables

**Figure 1 sensors-22-02629-f001:**
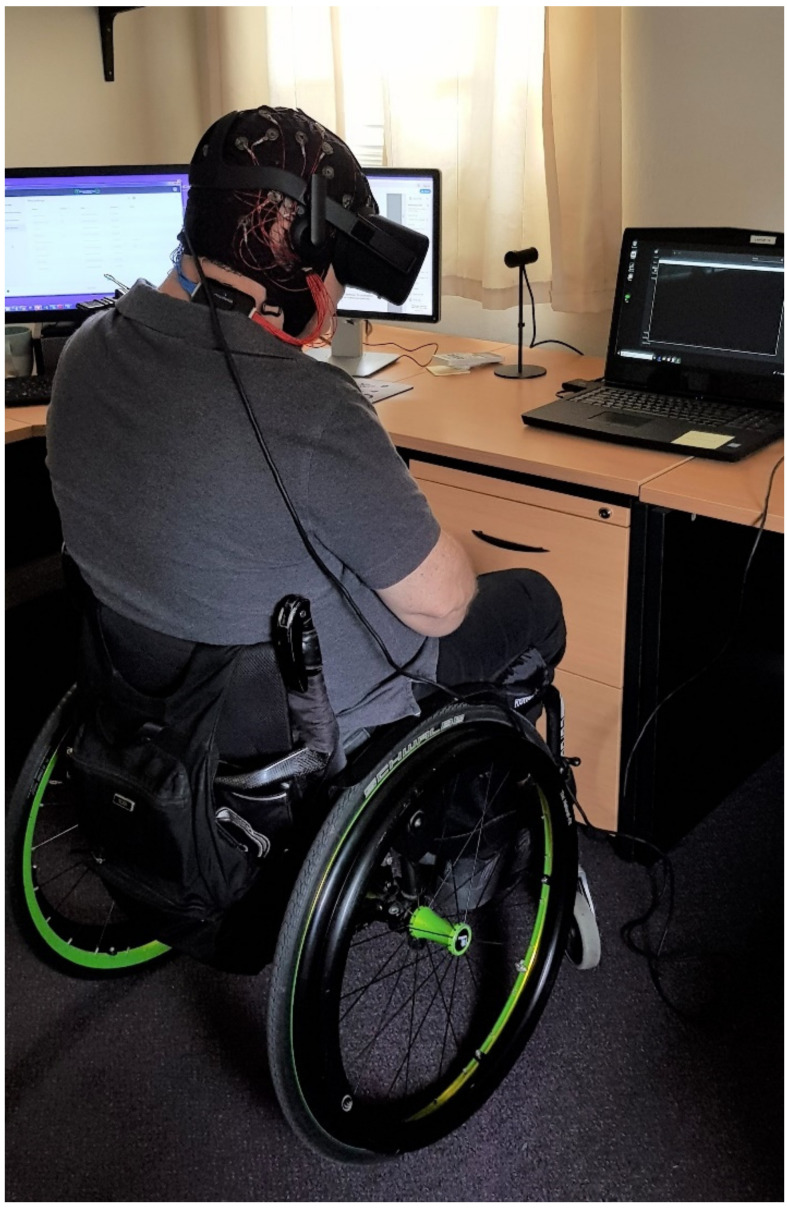
Participant with EEG and 3D HMD VR set up.

**Figure 2 sensors-22-02629-f002:**
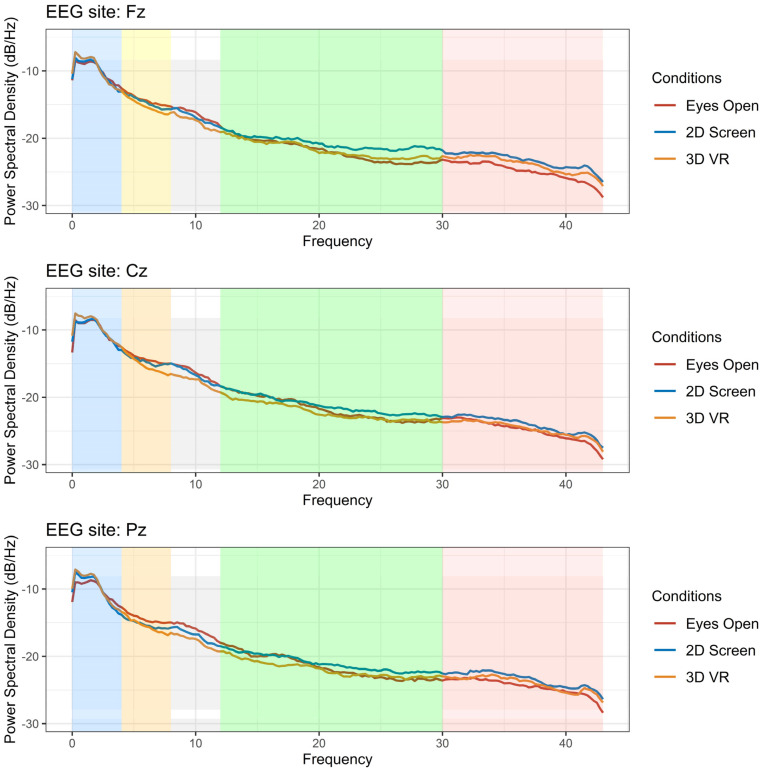
EEG average power spectrum for *n* = 15 participant during eyes-open task (red line), 2D screen (blue line) and 3D VR (orange line) in the Fz, Cz and Pz channels. Blue shade = Delta (1–4 Hz), Yellow shade = Theta (4–8 Hz), Gray shade = Alpha (8–12 Hz), Green shade = Beta (12–30 Hz) and Pink shade = Gamma (30–45 Hz).

**Figure 3 sensors-22-02629-f003:**
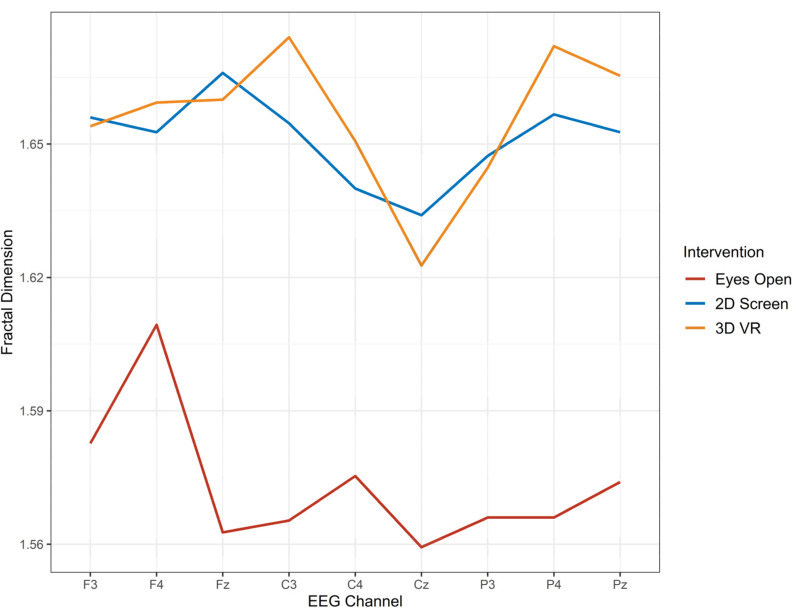
The fractal dimension from EEG activity during three conditions, no-eyes-open task (red), 2D screen task (blue) and 3D VR task (orange).

**Table 1 sensors-22-02629-t001:** Participant demographic characteristics (*n* = 15).

Characteristics		*n* (%)
Age	Mean (SD)	56.0 (13.1)
Years in pain since injury	Mean (SD)	20.0 (12.3)
ASIA impairment grade	A	10 (66.7)
	B	2 (13.3)
	C	0 (0.0)
	D	3 (20.0)
Injury level	Paraplegia	13 (86.7)
	Tetraplegia	2 (13.3)
Pain consistency	Constant	12 (80.0)
	Intermittent	3 (20.0)
Pain medication	Multiple	4 (26.7)
	Single	6 (40)
	None	5 (33.3)

**Table 2 sensors-22-02629-t002:** Relative EEG delta power for the three conditions EO-no task, 2D screen and 3D VR.

	Three Conditions			Main Effect between Conditions		
Channel	EO-No Task Mean (SE)	2D Screen Mean (SE)	3D VR Mean (SE)	F (2, 28)	*p*	η^2^_P_
F3	0.370 (0.018)	0.338 (0.011)	0.422 (0.010)	8.852	0.001	0.387
F4	0.358 (0.018)	0.355 (0.019)	0.397 (0.015)	2.674	0.087	0.160
Fz	0.366 (0.019)	0.349 (0.013)	0.365 (0.015)	0.373	0.692	0.026
C3	0.325 (0.016)	0.341 (0.017)	0.374 (0.019)	2.628	0.090	0.158
C4	0.329 (0.023)	0.342 (0.014)	0.352 (0.018)	0.539	0.589	0.037
Cz	0.348 (0.014)	0.363 (0.018)	0.375 (0.018)	0.769	0.473	0.052
P3	0.346 (0.013)	0.326 (0.011)	0.359 (0.019)	1.487	0.243	0.096
P4	0.322 (0.013)	0.344 (0.013)	0.338 (0.019)	0.525	0.597	0.036
Pz	0.352 (0.015)	0.365 (0.015)	0.388 (0.016)	2.051	0.147	0.128

SE = Standard error, η^2^_P_ = Partial eta squared.

**Table 3 sensors-22-02629-t003:** Relative EEG theta power for the three conditions EO-no task, 2D screen and 3D VR.

	Three Conditions			Main Effect between Conditions		
Channel	EO-No Task Mean (SD)	2D Screen Mean (SD)	3D VR Mean (SD)	F (2, 28)	*p*	η^2^_P_
F3	0.169 (0.009)	0.166 (0.010)	0.132 (0.013)	8.031	0.002	0.365
F4	0.180 (0.013)	0.159 (0.012)	0.149 (0.013)	2.507	0.100	0.152
Fz	0.192 (0.011)	0.165 (0.012)	0.155 (0.011)	3.165	0.058	0.184
C3	0.191 (0.010)	0.167 (0.013)	0.145 (0.011)	5.135	0.013	0.268
C4	0.191 (0.012)	0.152 (0.012)	0.172 (0.011)	4.261	0.024	0.233
Cz	0.196 (0.008)	0.173 (0.014)	0.172 (0.013)	2.214	0.128	0.137
P3	0.203 (0.009)	0.172 (0.009)	0.164 (0.012)	8.059	0.002	0.365
P4	0.201 (0.011)	0.158 (0.011)	0.148 (0.012)	10.484	<0.001	0.428
Pz	0.180 (0.007)	0.147 (0.011)	0.148 (0.014)	4.274	0.024	0.234

SE = Standard error, η^2^_P_ = Partial eta squared.

**Table 4 sensors-22-02629-t004:** Relative EEG alpha power for the three conditions EO-no task, 2D screen and 3D VR.

	Three Conditions			Main Effect between Conditions		
Channel	EO-No Task Mean (SD)	2D Screen Mean (SD)	3D VR Mean (SD)	F (2, 28)	*p*	η^2^_P_
F3	0.126 (0.013)	0.111 (0.009)	0.074 (0.006)	10.625	<0.001	0.431
F4	0.115 (0.009)	0.107 (0.014)	0.081 (0.008)	6.952	0.004	0.332
Fz	0.114 (0.009)	0.098 (0.009)	0.088 (0.003)	3.056	0.063	0.179
C3	0.139 (0.015)	0.109 (0.011)	0.090 (0.008)	4.490	0.020	0.243
C4	0.146 (0.018)	0.117 (0.012)	0.106 (0.010)	4.764	0.017	0.254
Cz	0.132 (0.013)	0.109 (0.012)	0.096 (0.008)	4.520	0.020	0.244
P3	0.128 (0.012)	0.112 (0.010)	0.102 (0.012)	1.501	0.240	0.097
P4	0.144 (0.014)	0.101 (0.009)	0.100 (0.007)	5.897	0.007	0.296
Pz	0.127 (0.010)	0.099 (0.008)	0.089 (0.011)	6.715	0.004	0.324

SE = Standard error, η^2^_P_ = Partial eta squared.

**Table 5 sensors-22-02629-t005:** Relative EEG beta power for the three conditions EO-no task, 2D screen and 3D VR.

	Three Conditions			Main Effect between Conditions		
Channel	EO-No Task Mean (SD)	2D Screen Mean (SD)	3D VR Mean (SD)	F (2, 28)	*p*	η^2^_P_
F3	0.157 (0.012)	0.175 (0.012)	0.138 (0.007)	3.583	0.041	0.204
F4	0.166 (0.013)	0.175 (0.012)	0.148 (0.013)	1.818	0.181	0.115
Fz	0.157 (0.013)	0.184 (0.012)	0.160 (0.013)	1.835	0.178	0.116
C3	0.186 (0.011)	0.188 (0.009)	0.174 (0.016)	0.598	0.557	0.041
C4	0.174 (0.012)	0.180 (0.010)	0.183 (0.014)	0.300	0.743	0.021
Cz	0.166 (0.008)	0.173 (0.013)	0.161 (0.013)	0.325	0.725	0.023
P3	0.158 (0.008)	0.190 (0.008)	0.175 (0.012)	4.022	0.029	0.223
P4	0.181 (0.009)	0.181 (0.011)	0.199 (0.021)	0.673	0.518	0.046
Pz	0.169 (0.010)	0.179 (0.011)	0.150 (0.012)	2.251	0.124	0.139

SE = Standard error, η^2^_P_ = Partial eta squared.

**Table 6 sensors-22-02629-t006:** Relative EEG gamma power for the three conditions EO-no task, 2D screen and 3D VR.

	Three Conditions			Main Effect between Conditions		
Channel	EO-No Task Mean (SD)	2D Screen Mean (SD)	3D VR Mean (SD)	F (2, 28)	*p*	η^2^_P_
F3	0.058 (0.007)	0.085 (0.009)	0.061 (0.005)	7.584	0.002	0.351
F4	0.066 (0.008)	0.082 (0.008)	0.066 (0.007)	1.953	0.161	0.122
Fz	0.060 (0.008)	0.082 (0.007)	0.073 (0.008)	2.798	0.078	0.167
C3	0.061 (0.005)	0.082 (0.008)	0.074 (0.007)	3.904	0.032	0.218
C4	0.061 (0.009)	0.076 (0.008)	0.077 (0.010)	1.323	0.283	0.086
Cz	0.056 (0.005)	0.073 (0.10)	0.066 (0.007)	1.884	0.171	0.119
P3	0.060 (0.006)	0.089 (0.007)	0.078 (0.009)	9.533	<0.001	0.405
P4	0.059 (0.006)	0.085 (0.009)	0.086 (0.009)	7.017	0.003	0.334
Pz	0.064 (0.007)	0.080 (0.009)	0.068 (0.008)	2.108	0.140	0.131

SE = Standard error, η^2^_P_ = Partial eta squared.

**Table 7 sensors-22-02629-t007:** Fractal dimensions in nine EEG sites for the three conditions EO-no task, 2D screen and 3D VR.

	Three Conditions			% Change from EO		Main Effect	
Channel	EO-No Task Mean (SD)	2D Screen Mean (SD)	3D VR Mean (SD)	2D screen	3D VR	F (2, 13)	*p*
F3	1.58 (0.09)	1.66 (0.11)	1.65 (0.12)	5.06	4.43	3.25	0.07
F4	1.61 (0.11)	1.65 (0.12)	1.66 (0.10)	2.48	3.11	1.45	0.27
Fz	1.56 (0.08)	1.67 (0.09)	1.66 (0.09)	7.05	6.41	9.87	0.002
C3	1.57 (0.06)	1.65 (0.09)	1.67 (0.08)	5.10	6.37	18.13	<0.001
C4	1.58 (0.10)	1.64 (0.09)	1.65 (0.09)	3.80	4.43	4.81	0.03
Cz	1.56 (0.09)	1.63 (0.11)	1.62 (0.09)	4.49	3.84	4.62	0.03
P3	1.58 (0.10)	1.64 (0.09)	1.65 (0.09)	3.80	4.43	4.81	0.03
P4	1.57 (0.11)	1.66 (0.09)	1.67 (0.09)	5.73	6.37	8.39	0.005
Pz	1.57 (0.09)	1.65 (0.10)	1.67 (0.12)	5.10	6.37	10.20	0.002

**Table 8 sensors-22-02629-t008:** Performance of binary classifications using ANN models.

Binary Classification	Sensitivity (%)	Specificity (%)	Accuracy (%)	ROC
3D VR vs. 2D screen + EO	43.8	94.1	80.3	0.767
3D VR vs. EO	78.0	79.5	78.8	0.877
3D VR vs. 2D screen	68.3	73.8	71.1	0.744
2D screen vs. EO	66.7	76.7	72.4	0.730

## Data Availability

The datasets generated and/or analysed during the current study are available from the corresponding author on reasonable request.

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
