# Peer review of "An Exploratory EEG Analysis on the Effects of Virtual Reality in People with Neuropathic Pain Following Spinal Cord Injury"

_sensors, 2022, doi:10.3390/s22072629_

Round 1

Reviewer 1 Report

Data from 15 adult males with chronic SCI and NP were analyzed. Participants underwent baseline EEG (eyes open, focused on blank computer screen), followed by 2D and 3D VR conditions in random order and separated by a 1-hr “washout” period. Participants were tested in a single session in a temperature-controlled room. The study was preregistered. The authors found reductions of NP ratings (on a 11-point scale) during VR conditions, associated with decreased alpha band (8-13 Hz) power and increased  brain complexity (FD) measures. 3D VR was different from both EO and 2D VR conditions. The authors suggest the results have implications for using VR applications as a therapeutic intervention for NP in people with SCI.

The study and results are interesting. For the results to have implications for therapeutic intervention, though, it will be important to know if the decrease in NP persists beyond the session and if so, for how long. If it lasts only during a session, then perhaps any attention-directing activity would be as effective? A related question regards the mechanisms underIying the effect: is there something specific about VR or is it simply a sophisticated attention-directing activity that provides distraction from NP? More discussion about the potential mechanisms would be helpful. Also, the mean pain level at baseline was not high (M = 4.9); would VR intervention be effective if patients had more severe pain? 

--Figure 3 would be more informative as a 3-panel figure, showing Cz and Pz as well as Fz.

--Tables summarizing statistics are good--but please add a column with effect size measures to each

--Please make the following grammar corrections:

P1L35 also is associated

L137 was modified

L148—sentence (“The 11-point NPRS…”) needs editing

L209 data were

L200 The FD for this study

L246 Tables show

L252 differences were also

L289 differences

L292 found differences between

L333 et al. (remove comma)

L340 Lim et al. [38]

L348 (2021) found [remove comma]

Reviewer 2 Report

The study aimed to determine if the use of VR was associated with changes in EEG patterns linked to the presence of neuropathic pain. The authors compared EEG patterns in three states, a resting eyes-opened with no task (EO-no task), while using a 2D screen-based VR (2D screen) as an active control and during immersive 3D HMD VR (3D VR). The comparison was done using Fourier analysis of energy in different EEG bands, and by comparing fractal dimensions of the EEG signal. The authors report evidence of reduced pain during immersion in VR environment.

  1. Please report in Sec. 2.4 that this part of the study was published earlier [36].
  2. Please format the FD equation (pg. 5, line 161) for easy viewing. Please also explain Gauss' notation.
  3. It would be nice to show the alpha, beta, delta, theta regions on Fig. 2.
  4. Please discuss why the sensitivity of binary classification is generally low (Table 7).
  5. Please discuss if the ANN binary classification results match with analysis of oscillatory and non-oscillatory waveform characteristics.
  6. Please indicate the % rise in FD for 2D/3D vs. EO (Fig. 6).
  7. It is desirable to include a follow up assessment to determine if any of 2D/3D VR interventions had lasting effect on pain suppression.
  8. There is a typo on pg. 6, line 249.

Reviewer 3 Report

In a within-subject, randomised cross-over pilot trial the authors investigated a novel therapeutic intervention using virtual reality head-mounted device for pain relief in neuropathic pain in people with spinal cord injury. The study is interesting and of potential relevance, yet the unclarities in the EEG methodology deem it difficult to judge the quality of the results (see my detailed comments below).

EEG pre-processing and analysis:

  1. The description of the pre-processing part must be improved. At present, it is unclear which pre-processing steps were undertaken and what was their order (i.e., after data filtering, was there identification of bad stretches and bad channels, that were followed by ICA correction?). Which algorithm was used for the ICA? What do the authors mean by stating that movement artifacts were removed by ICA?
  2. Did the VR-glasses not collide with the EEG-electrodes, i.e., could the signal of all EEG electrodes be collected? On a similar note, was there no interference in the EEG signal caused by the VR-device? Did the authors apply a notch filter? In the paper I only found information on the high-pass filter.
  3. Lines 166-167 - it is unclear to me why the authors used the given frequency ranges, e.g., delta is often defined as 1-4Hz or 0.5-4Hz. Please provide a refence that justifies the choice of the ranges of the given frequency bands. Why was not the gamma band examined? Depending on the task demands in the 2D screen and 3D VR conditions, which description I could not find in the paper, I would expect modulation in higher frequencies such as gamma.
  4. Why did the authors record 32-channels but used only 9 for the analysis? What is the rationale for using the given channels? Please clarify and provide corresponding references.
  5. I am not sure if I understood it correctly, was the baseline condition 1-minute long while the other two conditions 5-minute long? Was EEG collected during the VR- and 2D screen interventions? The length of the artifact-free EEG that was subject to analyses (i.e., after pre-processing) must be specified for each condition.
  6. The authors present the power spectrum in the Fz channel (Fig. 2), which they call representative. I would encourage the authors to also present topomaps for each condition of interest and each frequency band investigated. The major advantage of topomaps is that they show the distribution across the entire scalp.

2D screen and 3D VR conditions:

  1. I did not find a detailed description of the 2D screen and 3D VR conditions. Which screen size was used in the 2D screen condition, what were the sampling rates of the screen and the VR-glasses? What did the task entail? All these must be specified.

Minor comments:

  1. Baseline (eyes open) condition: How did the authors ensure that the participants were fixating the middle of the screen, did they present a fixation cross?
  2. Please specify the software in which the EEG data was acquired and pre-processed.
  3. Lines 155-156 the authors state: “(…) The EEG signals were re-sampled to 128Hz and re-referenced using an average reference from the 32 channels.” I guess the authors mean “transformed to the average reference”. Please specify how the average reference was calculated and indicate the recording reference used in the study.
  1. Lines 170-171 – this sentence is confusing. Do the authors mean the average of the 20-sec long time windows?
  2. What is the rationale for extracting 20-sec windows for the analysis?
  3. The authors state: “Participants were required to report any headset discomfort and cyber-sickness (includes symptoms of nausea, vomiting, headache, vertigo and fatigue) prior to, during or after using the 3D HMD VR device.” - but I did not find the results of thereof.

Reviewer 4 Report

It is an honor to read such a cutting-edge article.

A few comments:

  1. I feel that the photo in Figure 1 could be replaced with a better one. The current photos are unclear in terms of resolution and setup demonstration. The resolution of Figures 2 and 3 is also poor.
  2. Many formulas have typesetting problems. I don't know whether they should be solved by the typesetter or corrected by the authors themselves.
  3. The title and content of 2.6 do not seem to match exactly. The title can, for example, include "evaluation/metrics/criteria."
  4. As a researcher using ML models for biosignal research, I look forward to seeing a more detailed analysis of ANN results, not only in terms of listing or comparing accuracies and other indicators. The expression "can be classified with reasonable accuracy" as an analysis conclusion of an article is somewhat insufficient from the perspective of ML.

Reviewer 5 Report

Article : sensors-1647174

This manuscript describes very interesting findings in the VR applications as a therapeutic intervention for neuropathic pain in people with spinal cord injury. The subject of the manuscript is very consistent with the scope of the Journal and the paper adds new and original values to the Journal. The manuscript is well presented and it could be of interest to many readers.

Following revisions are suggested:

  1. Figures 2 and 3 have poor resolution. Please replace them.
  1. Give a detailed scheme of the setup.
  1. What are the difference or advantages of the proposed method compared with conventional methods?

Round 2

Reviewer 3 Report

The authors have thoroughly addressed my concerns. As for this review, there are only few remaining points, the first two of which I do not consider trivial.

Major points:
1. For a successful ICA signal decomposition the continous EEG data must be clean of artifacts. The authors state:
a. “6. With the 20 second EEG segment EEG artifact from blinks, electrode movements and cardiac signal components were analysed using independent component analysis (ICA) using EEGLab [35].” - What do the authors mean by “electrode movements”? Do they mean “channels containing artifacts”? If so, these must be excluded before applying any ICA algorithm. ICA algorithms can deal with artifacts such as e.g., eye blinks, saccades, cardiac artifact, or even muscle artifact, yet they cannot compensate for noisy channels. Bad or noisy channels are typically excluded prior to ICA and may be interpolated in a later step. This becomes even more critical for the type of analyses that the authors conduct and that strictly require a clean EEG signal for the analyses to be meaningful. 

b. On a similar note, in both 2D screen and 3D VR conditions I would expect rather large lateral eye movement artifacts, which are typically well detected by the available ICA algorithms, did the authors use ICA to correct for these?

2. Regarding the use of average reference:
I am not sure why the authors decided to apply the average reference and why they decided to apply it before the artifact correction. When applying the so-called Common Average Reference, the new reference is the average electrical activity measured across all scalp channels. Re-referencing is achieved by creating an average of all scalp channels and subtracting the resulting signal from each channel. After re-referencing, the overall electrical activity across all channels will sum up to zero at each time point. When using this reference, amplitudes will be overall reduced, but each channel will contribute equally to the new reference. So before doing this we would typically make sure that the EEG is clean of artifacts.
Moreover, the re-referenced signal should not be biased towards any specific location on the scalp. However, the head is typically not evenly covered by electrodes. Instead, electrodes are often placed more densely on the top while being placed less densely in the lower parts of the head. If the signal on top of the head is overrepresented in the reference, the amplitudes in this region will be reduced. To minimize this bias, the average reference typically requires a higher coverage with at least 64 up to 256 electrodes. Optimally, electrodes should thereby be evenly spaced and cover over 50% of the head surface. 

Minor points:
3. Line 185 - I belive the authors mean „with a 0.1 Hz high-pass filter“

4. Line 359: What do the authors mean by the following sentence?: „Increases in beta activity occurred only at higher frequencies in the gamma frequencies (30-45 Hz).“

5. Figure 2. Eyes open, 2D screen and 3D VR - all together are rather conditions than interventions.
